# Assessment of the Correlation and Diagnostic Accuracy between Cerebrospinal Fluid and Plasma Alzheimer’s Disease Biomarkers: A Comparison of the Lumipulse and Simoa Platforms

**DOI:** 10.3390/ijms25094594

**Published:** 2024-04-23

**Authors:** Farida Dakterzada, Raffaela Cipriani, Ricard López-Ortega, Alfonso Arias, Iolanda Riba-Llena, Maria Ruiz-Julián, Raquel Huerto, Nuria Tahan, Carlos Matute, Estibaliz Capetillo-Zarate, Gerard Piñol-Ripoll

**Affiliations:** 1Cognitive Disorders Unit, Cognition and Behaviour Study Group, Santa Maria University Hospital, IRBLleida, 25198 Lleida, Spain; fdakterzada@irblleida.cat (F.D.); aarias@gss.cat (A.A.); yriba@gss.cat (I.R.-L.); mruizj@gss.cat (M.R.-J.); rhuerto@gss.cat (R.H.); ntahan@gss.cat (N.T.); 2Achucarro Basque Center for Neuroscience, 48940 Leioa, Spain; raffaela.cipriani@achucarro.org (R.C.); carlos.matute@ehu.eus (C.M.); estibaliz.capetillo@ehu.eus (E.C.-Z.); 3Laboratori ClínicInstitut Català de la Salut (ICS), Hospital Universitari Arnau de Vilanova, 25198 Lleida, Spain; relopez.lleida.ics@gencat.cat; 4Department of Neurosciences, Faculty of Medicine and Nursery, University of the Basque Country (UPV/EHU), 48940 Leioa, Spain; 5CIBERNED, Centro de Investigación Biomédica en Red Enfermedades Neurodegenerativas, 28029 Madrid, Spain; 6Department of Neurosciences, Faculty of Pharmacy, University of the Basque Country (UPV/EHU), 01008 Vitoria-Gasteiz, Spain; 7IKERBASQUE, Basque Foundation for Science, 48009 Bilbao, Spain; 8Departament de Medicina Experimental, Facultat de Medicina, Universitat de Lleida (UDL), 25002 Lleida, Spain

**Keywords:** Alzheimer’s disease, biomarker, plasma, Simoa, Lumipulse, cerebrospinal fluid, cut-off

## Abstract

We compared the clinical and analytical performance of Alzheimer’s disease (AD) plasma biomarkers measured using the single-molecule array (Simoa) and Lumipulse platforms. We quantified the plasma levels of amyloid beta 42 (Aβ42), Aβ40, phosphorylated tau (Ptau181), and total tau biomarkers in 81 patients with mild cognitive impairment (MCI), 30 with AD, and 16 with non-AD dementia. We found a strong correlation between the Simoa and Lumipulse methods. Concerning the clinical diagnosis, Simoa Ptau181/Aβ42 (AUC 0.739, 95% CI 0.592–0.887) and Lumipulse Aβ42 and Ptau181/Aβ42 (AUC 0.735, 95% CI 0.589–0.882 and AUC 0.733, 95% CI 0.567–0.900) had the highest discriminating power. However, their power was significantly lower than that of CSF Aβ42/Aβ40, as measured by Lumipulse (AUC 0.879, 95% CI 0.766–0.992). Simoa Ptau181 and Lumipulse Ptau181/Aβ42 were the markers most consistent with the CSF Aβ42/Aβ40 status (AUC 0.801, 95% CI 0.712–0.890 vs. AUC 0.870, 95% CI 0.806–0.934, respectively) at the ≥2.127 and ≥0.084 cut-offs, respectively. The performance of the Simoa and Lumipulse plasma AD assays is weaker than that of CSF AD biomarkers. At present, the analysed AD plasma biomarkers may be useful for screening to reduce the number of lumbar punctures in the clinical setting.

## 1. Introduction

Alzheimer’s disease (AD) is the most common type of dementia, and its incidence is expected to increase in the coming years. AD-specific neuropathology consists of extracellular amyloid plaques arising from the accumulation of amyloid beta protein (Aβ, A) and intracellular neurofibrillary tangles formed by aggregations of hyperphosphorylated tau protein (Ptau, T) [1], while neurodegeneration (N), the third neuropathological aspect of AD, is a nonspecific hallmark and can be caused by several neurodegenerative diseases. Regarding Ptau in AD, the phosphorylation at position threonine 181 (Ptau181) is the most thoroughly examined Ptau epitope [2]. Currently, monitoring ATN pathologies in cerebrospinal fluid (CSF) (i.e., via the quantification of Aβ42 or Aβ42/Aβ40 for A, Ptau for T, and total tau (Ttau) or neurofilament light (NfL) for N) or by imaging techniques is an established way to more accurately diagnose AD and select AD patients for clinical trials [2]. However, obtaining CSF is invasive, which limits its use for concurrent monitoring of therapeutic trials and drug efficacy and for longitudinal studies where multiple lumbar punctures are needed. In addition, the high cost of imaging techniques limits their routine use in clinical practice, clinical trials, and research studies. Therefore, the use of blood-based biomarkers is desirable because of the minimal invasiveness and cost effectiveness of these methods.

Plasma biomarkers of AD have long been unavailable because of the detection limit of the available immunoassay methods (picomolar concentration, 10^−12^ M). However, recent technological advances have led to increased opportunities to measure biomarkers in blood. Among these methods, the single-molecule array (Simoa) method is the most established. It can detect proteins in plasma or serum at subfemtomolar (<10^−15^ M) concentrations. The detection system is based on capturing the analyte by target-specific antibodies coupled to paramagnetic particles. This immune complex is confined to femtoliter-sized wells, which restricts the diffusion of the signal and increases the sensitivity. Another platform that has also developed kits for the measurement of plasma AD biomarkers (Aβ42, Aβ40, and Ptau) is Lumipulse (Fujirebio Europe NV, Gent, Belgium). In this platform, CSF and plasma biomarkers are measured using the chemiluminescent enzyme immunoassay (CLEIA) method. CSF biomarkers measured using this platform have demonstrated good concordance with Aβ-PET and CSF Aβ42 status determined by ELISA [3,4]. The detection limit for both plasma and CSF AD biomarkers measured by the Lumipulse platform is at picomolar concentrations.

Several recent studies have assessed the performance of plasma biomarkers quantified using Simoa technology for distinguishing AD brain pathology status [5,6] and Aβ-PET status [7,8,9], determining diagnostic accuracy [6,10], monitoring cognitive changes [5,6,9,11], and performing differential diagnosis [5,12]. However, the performance of plasma AD biomarkers determined using Lumipulse has been little studied [13], and comparisons of the Simoa and Lumipulse platforms need further research as these two platforms were previously compared only regarding Ptau181 performance [14,15]. Therefore, the objectives of the present study were as follows: (a) to assess the correlation and concordance between plasma AD biomarkers measured using the Lumipulse and Simoa methods; (b) to define which plasma biomarker in each platform has a better correlation with CSF Aβ42/40 status measured by Lumipulse; (c) to evaluate the diagnostic accuracy in discriminating between AD from other non-AD dementia plasma AD biomarkers measured by each method and their comparison with the diagnostic accuracy of the Lumipulse CSF AD biomarker with best discriminating power; (d) to determine the cut-offs of plasma AD biomarkers measured by Simoa and Lumipulse based on the best discriminating accuracy between the CSF Aβ42/40 positive and negative status; and (e) to evaluate the diagnostic accuracy of the ATN classification with Simoa.

## 2. Results

### 2.1. Study Population

The characteristics of the study population, including demographic data, comorbidities, MMSE score, *APOE4* status, and CSF and plasma levels of AD biomarkers, are summarized in Table 1. The average age of the participants was 74 (6.7 SD) years, and 55.9% were female. The diagnoses in the cohort were as follows: 30 (23.6%) had AD, 81 (63.7%) had MCI, and 16 (12.6%) had non-AD dementia. There were no significant differences between diagnostic groups for demographic data or comorbidities; however, the groups differed with respect to the MMSE score or the frequency of the *APOE ε4* genotype (*p* < 0.001 and *p* = 0.001, respectively). The Lumipulse CSF Aβ42, Ttau, and Ptau181 concentrations and the Aβ42/40, Ptau181/Aβ42, and Ttau/Aβ42 ratios were significantly different among the three diagnostic groups. The Lumipulse and Simoa levels were significantly different between the two groups for plasma Ptau181 (*p* = 0.009 and *p* = 0.011, respectively) and for the plasma Ptau181/Aβ42 ratio (*p* < 0.001 and *p* = 0.010, respectively). In addition, the Lumipulse test revealed a significant difference in the plasma Aβ42 concentration (*p* = 0.04) (Table 1).

### 2.2. Correlations and Concordance between the Plasma Levels of AD Biomarkers Measured by the Simoa and Lumipulse Platforms

To evaluate correlations between plasma AD biomarkers measured by either the Lumipulse or Simoa platform, we used Pearson’s correlation. The correlation coefficient (r) between these two platforms was 0.794 for Aβ42 (*p* < 0.001) (Figure 1A), 0.891 for Ptau181 (*p* < 0.001) (Figure 1C), 0.572 for Aβ42/40 (*p* < 0.001) (Figure 1E), and 0.837 for Ptau181/Aβ42 (*p* < 0.001) (Figure 1G). These results indicated that although there was a moderate correlation regarding Aβ42/40, the two methods had a high correlation between them for Aβ42, Ptau181, and Ptau181/Aβ42.

To assess the concordance between the Simoa and Lumipulse tests concerning the values of the plasma AD biomarkers, paired sample t tests and Bland–Altman plots were used. Regarding Aβ42, we observed a proportional systematic bias of −16.157 (*p* < 0.001) (Figure 1B). This means that on average, Lumipulse detected 16.157 pg/mL more Aβ42 than Simoa. The regression line demonstrated a proportional systematic bias, with a negative trend of differences as the magnitude of Aβ42 increased. For Ptau181, the bias between methods was 0.359 (*p* < 0.001), indicating that, on average, Simoa detected 0.359 pg/mL more Ptau181 than Lumipulse (Figure 1D). The regression line for the differences indicated that there was a significant mild positive trend in the differences as the magnitude of the measured variable increased. For Aβ42/40, the mean difference was −0.0396 (*p* < 0.001) (Figure 1F). The regression line for the differences indicated that there was a systematic proportional bias between the values of the two methods, with a mild negative trend in the differences as the magnitude of the Aβ42/40 values increased. Finally, regarding the Ptau181/Aβ42 ratio, there was a systematic proportional bias of 0.263 units between the methods (Figure 1H). The regression line indicated a positive trend in differences as the magnitude of the Ptau181/Aβ42 ratio increased. For all assays evaluated, approximately 95% of the measured values were within ±1.96 SD of the bias. These results indicated a lack of concordance with respect to all measured biomarkers between the Simoa and Lumipulse platforms.

### 2.3. Correlation between the Lumipulse CSF Levels of AD Biomarkers and Their Plasma Levels Measured by Simoa and Lumipulse

Next, we evaluated the correlations between CSF AD biomarkers measured by the Lumipulse platform and their equivalent plasma levels measured by either the Lumipulse or Simoa platform. The correlation coefficients between AD biomarkers in CSF measured by Lumipulse and the same markers quantified in plasma by Simoa were 0.211 (*p* = 0.019) for Aβ42/40, 0.346 (*p* < 0.001) for Ptau181/Aβ42, and 0.363 (*p* < 0.001) for Ptau181 (Figure 2, Appendix A). Furthermore, Simoa Ptau181 correlated significantly with CSF Aβ42 (*r* = −0.269, *p* = 0.002) and CSF Aβ42/40 (*r* = −0.400, *p* < 0.001). In contrast, the correlations for the Aβ42, Aβ40, Ttau, and Ttau/Aβ42 biomarkers were not significant (Appendix A).

In agreement with the results of Simoa, the correlations between CSF measurements of AD biomarkers and plasma levels of these biomarkers quantified by the Lumipulse platform indicated weak but significant correlations for Aβ42/40 (*r* = 0.423, *p* < 0.001), Ptau181/Aβ42 (*r* = 0.332, *p* < 0.001) and Ptau181 (*r* = 0.330, *p* < 0.001) (Figure 3, Appendix A). In addition, the Lumipulse platform showed a weak but significant correlation with Aβ42 (*r* = 0.251, *p* = 0.005) (Figure 3). As we also showed with Simoa, Lumipulse Ptau181 correlated significantly with CSF Aβ42 (*r* = −0.269, *p* = 0.011) and CSF Aβ42/40 (*r* = −0.350, *p* < 0.001) (Appendix A).

Correlations with plasma and CSF Ttau levels were not analysed with Lumipulse due to the lack of a Ttau assay for this platform.

### 2.4. Diagnostic Accuracy of the Plasma Biomarkers Measured by Simoa and Lumipulse in Comparison to the Diagnostic Accuracy of CSF Biomarkers Measured by Lumipulse

The discriminating power of plasma biomarkers quantified using Simoa and Lumipulse with respect to the diagnosis of AD versus non-AD dementia was evaluated using binary logistic regression. We first evaluated the diagnostic accuracy of Lumipulse CSF AD biomarkers to compare the statistical results of plasma biomarkers with the CSF AD biomarker with best diagnostic accuracy and to confirm the previously reported data [3]. Among the CSF AD biomarkers and ratios, Aβ42/40 was the variable with the best discriminating power (AUC 0.879 95% CI 0.766–0.992), as we showed previously [3] (Table 2). This variable had 87.9% sensitivity and 75% specificity for correctly classifying diagnostic groups, and its predictive accuracy was estimated to be 89.1%. Among the plasma biomarkers measured by Simoa, Ptau181/Aβ42 had the best discriminating power (AUC 0.739 95% CI 0.592–0.887). The sensitivity and specificity for correctly classifying patients according to this variable were 82.8% and 56.3%, respectively. The total accuracy for Ptau181/Aβ42 was 73.3%. The Hanley–McNeil test demonstrated that the AUC of Simoa Ptau181/Aβ42 significantly differed from the AUC of CSF Aβ42/40 (z = 2.016, |z| > 1.96). Among the plasma biomarkers measured by the Lumipulse tool, Aβ42 performed better than the other biomarkers for discriminating between the diagnostic groups (AUC 0.735 95% CI 0.589–0.882). The sensitivity and specificity of this assessment according to Aβ42 were 73.3% and 56.3%, respectively, and the total accuracy was 67.4%. The AUC of the Lumipulse Aβ42 concentration was not significantly different from the AUC of the CSF Aβ42/40 concentration (z = 1.620, |z| > 1.96), as assessed using the Hanley and McNeil methods (Table 2). In addition, Lumipulse Ptau181/Aβ42 yielded a similar diagnostic accuracy as did Lumipulse Aβ42 (AUC 0.733, 95% CI 0.567–0.900) (Table 2). In summary, the Ptau181/Aβ42 ratio showed high diagnostic accuracy for discriminating AD patients from non-AD patients according to both the Lumipulse and Simoa platforms.

### 2.5. Plasma Biomarker Cut-Offs Based on the CSF Aβ42/40 Ratio Status

In the next step, to determine the cut-off values for plasma AD biomarkers measured by Simoa and Lumipulse, the CSF Aβ42/40 ratio measured by Lumipulse was selected as a reference. The cut-off for each biomarker or ratio was established to be the value that optimized the concordance with the CSF Aβ42/40-positive and Aβ42/40-negative status. First, we determined the cut-off values of CSF AD biomarkers for our study population (Table 3); these values were very close to the cut-off values defined by the manufacturer (Fujireibio) and to the cut-off values we previously reported [3]. Among the biomarkers measured by Simoa, Ptau181, and Ptau181/Aβ42 had good discriminating accuracy between the CSF Aβ42/40-positive and Aβ42/40-negative status (AUC 0.801, OPA (overall percent agreement) 81.6% and AUC 0.789, OPA 78.9%, respectively) at cut-off values of ≥2.127 and ≥0.270, respectively (Table 3, Figure 4). Among the Lumipulse plasma biomarkers, Ptau181 (AUC 0.810, OPA 79.4%), Aβ42/40 (AUC 0.813, OPA 71.5%), and Ptau181/Aβ42 (AUC 0.870, OPA 84.7%) had AUCs greater than 0.80 at cut-offs of ≥2.070, ≤0.076, and ≥0.084, respectively (Table 3, Figure 4).

### 2.6. Diagnostic Accuracy of the ATN Classification with the Simoa Platform

Finally, we evaluated the diagnostic accuracy of the ATN (A refers to the plasma Aβ42 (Aβ42/40) concentration, T refers to the plasma Ptau concentration, and N refers to the plasma Ttau concentration) classification with the Simoa platform. This analysis was not performed on the Lumipulse platform because the plasma level of Ttau cannot be measured with this platform. The study population was divided into 6 ATN (0, 1, 2, 3, 4, and 5) groups based on the cut-offs defined previously for the three core AD biomarkers [16]. Participants with ATN 0 were negative for all three biomarkers. Those with an ATN of 1 were positive for only Aβ42 or for only the Aβ42/40 ratio. ATN 2 participants were positive for Aβ42 or for the Aβ42/40 ratio and for Ptau. ATN 3 patients were positive for all three biomarkers. ATN 4 patients were positive for Aβ42 or for the Aβ42/40 ratio and Ttau. Finally, ATN 5 patients were negative for Aβ42 or the Aβ42/40 ratio but positive for Ptau, Ttau or both biomarkers. We determined the discriminating power of two ATN classifications for Simoa, one based on the results of Aβ42, Ttau, and Ptau181 and the other based on the Aβ42/40, Ttau, and Ptau181 values (Table 4). These results were compared with those from the CSF ATN classification based on Aβ42/40, Ptau181, and Ttau, as this classification had greater discriminating power than did the combination of Aβ42, Ptau181, and Ttau in this study (Table 4) and according to previous observations [3]. Our results indicated that the Simoa ATN classification based on Aβ42/40 (AUC 0.733, 95% CI 0.576–0.890) or Aβ42 (AUC 0.726, 95% CI 0.573–0.880) did not significantly differ from the CSF ATN analysis based on Aβ42/40 (AUC 0.802, 95% CI 0.639–0.965) after comparing AUCs with those of the Hanley and McNeil methods (|z| < 1.96) (Table 4). In conclusion, our results indicate that plasma biomarkers measured by Simoa and CSF biomarkers measured by Lumipulse have similar diagnostic accuracy based on both ATN classifications.

## 3. Discussion

We observed a high correlation but lack of concordance between plasma AD biomarkers measured with both the Simoa and Lumipulse platforms. Both platforms identified the P181/Aβ42 ratio as a good plasma biomarker for discriminating between AD patients and non-AD dementia patients and between patients with a positive and negative CSF Aβ42/40 ratio. However, our results also showed a lack of correlation between CSF measurements of AD biomarkers quantified using Lumipulse and plasma levels quantified using the Simoa and Lumipulse platforms in a cohort of patients with AD, MCI, or non-AD dementia. In addition, compared with CSF AD biomarkers, plasma biomarkers had a lower diagnostic accuracy for discriminating AD patients from non-AD patients. Finally, the diagnostic accuracy of the Simoa ATN classification was not significantly different from that of the CSF ATN classification.

We started by comparing the Simoa and Lumipulse platforms. The strong correlation between the Simoa and the Lumipulse indices across all the AD assays demonstrated that both platforms could detect these biomarkers with similar efficiency. The lack of concordance for all measured biomarkers between the Simoa and Lumipulse platforms may be due to the different methodologies used by each platform for quantification. In addition, the antibodies used for the quantification of AD biomarkers are not the same for these two methods. For the detection of Aβ42, Simoa uses clones H31L21 and 6E10 as capture and detection antibodies, respectively; Lumipulse uses clones 21F12 and 3D6 as capture and detection antibodies, respectively. This could explain the difference between the median concentration of plasma Aβ42 detected by the Lumipulse device and that measured by the Simoa device.

Once we compared the plasma AD biomarkers with the Simoa and Lumipulse platforms, we proceeded with the correlation with CSF AD biomarkers. The CSF biomarkers were quantified using Lumipulse platform as part of the routine clinical practice of our memory clinic. Consistently with the findings of several previous reports, we found a lack of correlation or weak correlation between the plasma and CSF levels of biomarkers [8,17]. These data may indicate that the plasma levels of AD biomarkers might be affected not only by the magnitude of brain pathology but also by systemic conditions. In fact, vascular disease conditions, such as white matter lesions, cerebral microbleeds, hypertension, diabetes, and ischaemic heart disease, can increase plasma Aβ42 and Aβ42/Aβ40 levels [18]. In addition, recent studies suggest that blood Ttau originates principally from systemic, nonbrain sources, as it is present in peripheral organs, such as the liver, kidney, and heart [19,20].

We also examined the ability of plasma AD biomarkers and their ratios to distinguish AD patients from non-AD patients. For both the Simoa and Lumipulse platforms, Ptau181/Aβ42 (AUC 0.739 for Simoa and 0.733 for Lumipulse) performed better than the other biomarkers in differentiating AD from non-AD dementia. These results complement those of a previously reported study indicating that the plasma Ptau181/Aβ42 concentration can predict both amyloid-PET and cognitive decline [9]. In the case of Lumipulse, plasma Aβ42 (AUC 0.735) also showed similar diagnostic accuracy as Ptau181/Aβ42. However, the diagnostic accuracy of the plasma biomarkers was significantly lower than the discriminating power of Lumipulse CSF Aβ42/40 (AUC 0.879), which is the biomarker with the most diagnostic accuracy and concordance with amyloid PET [3,21].

We used the CSF Aβ42/40 status for the determination of plasma marker cut-off values. This marker was shown to function as well as the amyloid PET visual read or to have diagnostic accuracy for the determination of AD CSF biomarker cut-offs in previous studies [3]. CSF Aβ42/40 is resistant to preanalytical variations [21,22], and it probably accounts for interindividual variability in overall Aβ production and CSF turnover [4]. Based on our results, for the Simoa platform, plasma Ptau181 (AUC 0.801) and Ptau181/Aβ42 (AUC 0.789) performed better than other plasma biomarkers or ratios in discriminating the Aβ42/40-positive or Aβ42/40-negative status. These results are consistent with the results of previous studies in which Simoa plasma Ptau181 or Ptau181/Aβ42 was shown to be associated with an increase in Aβ deposition measured using Aβ PET in cognitively unimpaired adults [23] and patients with AD [9,24], as previously mentioned. In addition, Ptau181 has shown the strongest overall sensitivity and specificity for detecting neuropathological changes in AD [5]. Furthermore, Ptau181 and Ptau181/Aβ42 have been reported to be better than other plasma markers at predicting dementia risk [17] and the rate of cognitive decline [5,9].

Consistently with the results from Simoa, Lumipulse Ptau181 and Ptau181/Aβ42 also performed well in discriminating Aβ42/40-positive and Aβ42/40-negative status (AUC 0.810 and 0.870, respectively). Additionally, the plasma Aβ42/40 concentration measured by Lumipulse exhibited good discriminating power (AUC 0.813) in contrast to the Simoa Aβ42/40 concentration (AUC 0.641). This may be due to the use of the Lumipulse platform for the determination of both plasma and CSF Aβ42 and Aβ40 levels.

Finally, we also assessed the discriminating power of the ATN groups generated by the results of plasma Aβ42 (Aβ42/40), Ptau181, and Ttau from the Simoa platform. These results were compared with those of CSF ATNs based on Aβ42/40. The ATN classification provides a biological rather than a clinical definition of AD and was proposed by the NIAA research framework [2]. We found that the use of plasma Aβ42/40 instead of Aβ42 improved the accuracy of the ATN classification (AUC 0.733 vs. AUC 0.726), although this difference was not significant. In addition, the diagnostic accuracy of this ATN classification strategy was not significantly different from that of CSF-based ATN classification.

This study has several limitations. First, our study population lacked healthy control individuals. Second, our data regarding the diagnostic accuracy of plasma biomarkers may have been affected by the small number of AD (*n* = 30) and non-AD dementia patients (*n* = 16); however, our results were in accordance with those of some previous studies with larger sample sizes [5,12]. Third, plasma Ttau cannot be measured with the Lumipulse platform, so some of the comparisons were not possible. Fourth, CSF biomarkers were measured with only the Lumipulse platform. Fifth, no data regarding treatment were collected for the study population.

The main strength of our study is that we compared the clinical and analytical performance of the fully automated Simoa and Lumipulse platforms together in the same cohort of patients. In addition, our study population consisted of patients who attended a memory clinic; therefore, the study represents a more realistic application of plasma biomarkers in daily clinical practice. Importantly, both platforms identified Ptau181/Aβ42 as a good biomarker for discriminating AD from non-AD dementia, highlighting this ratio as an important biomarker for AD, as shown in previous publications [9].

We conclude that the clinical and analytical performance of both the Simoa and Lumipulse platforms for plasma analysis are comparable. Although the results obtained with plasma measured with the Simoa and Lumipulse platforms are promising, further investigations are needed. Currently, with these biomarkers, plasma cannot be a substitute for CSF as a diagnostic tool. However, AD plasma biomarkers can be useful for screening patients before performing a lumbar puncture.

## 4. Materials and Methods

### 4.1. Study Population

A total of 127 patients, including 30 AD, 81 mild cognitive impairment (MCI), and 16 non-AD dementia (including three vascular dementia, one semantic dementia, one tauopathy, two Lewy body dementia, two frontotemporal dementia, three behavioural variant frontotemporal dementia, one non-fluent progressive aphasia, two mixed dementia, and one unspecified dementia) patients, were included in this study. The study population was recruited consecutively between July 2018 and 2019 from patients attending the Cognitive Disorders Unit at the Hospital Universitari Santa Maria (Lleida, Spain). The inclusion criterion was presentation of suspected cognitive dysfunction for which the neurologist at the memory clinic requested CSF analysis. Therefore, patients with cognitive impairment caused by psychiatric problems or other conditions, such as stroke, brain tumour, and vitamin deficiency were excluded. The diagnosis of probable AD or MCI was performed based on the National Institute on Aging–Alzheimer’s Association (NIAA) criteria [25,26]. Each non-AD patient fulfilled the specific diagnostic criteria for the disorder considered (e.g., frontotemporal dementia, Lewy body dementia, etc.) [27,28,29]. Epidemiological data, including age, sex, education, and family history of cognitive impairment, were recorded using a structured interview conducted during the initial patient visit.

### 4.2. Sample Collection and Storage

CSF and plasma samples were collected between 8 a.m. and 10 a.m. after an overnight fast. CSF was collected in 10 mL polypropylene tubes (Sarstedt, Newton, NC, USA, 62.610.201). The tubes were inverted several times and centrifuged at 2000× *g* for 10 min at room temperature. The samples were aliquoted into two 2 mL polypropylene tubes (Sarstedt, 72.694.007), with each tube containing 1 mL of CSF. Blood samples were collected in EDTA-containing vacutainer tubes and centrifuged at 2000× *g* for 10 min at 4 °C to separate the plasma and buffy coat. All the samples were aliquoted and immediately stored at −80 °C until use. Samples were obtained with support from the IRBLleida Biobank, Lleida, Spain (B.0000682) and PLATAFORMA BIOBANCOS, Barcelona, Spain PT17/0015/0027 following the guidelines of Spanish legislation on this matter (Real Decreto 1716/2011).

### 4.3. Sample Analysis

The Lumipulse G600II automated platform (Fujirebio Europe NV, Gent, Brussels) was used to measure the CSF (3) and plasma Aβ42 levels (#230336 and 81301, respectively), Aβ40 levels (#231524 and 81298, respectively), Ptau181 levels (181P) (#230350 and 81288, respectively), and Ttau levels (only in CSF) (#230312). The following cut-offs for CSF biomarkers were determined by Fujirebio and used for data analysis: Aβ42 < 600 pg/mL, Aβ42/40 < 0.069, Ttau > 400 pg/mL, and Ptau181 > 56.5 pg/mL. The detection ranges of plasma Aβ40, Aβ42, and Ptau181 measured using Lumipulse were 0.10–5000 pg/mL, 0.10–1000 pg/mL, and 0.05–60.00 pg/mL, respectively.

A fully automated Simoa^®^ HD-1/HD-X Analyser (Quanterix, Billerica, MA, USA) was used for the quantification of plasma Aβ40, Aβ42, Ttau (neurology 3-Plex A (N3PA), #101995), and Ptau181 (#104111) using commercially available kits and according to the manufacturer’s instructions (Quanterix). The detection limits of the kits for Aβ40, Aβ42, Ttau, and Ptau181 were 0.196 (sample range of 0–600 pg/mL), 0.045 (sample range of 0–200 pg/mL), 0.019 (sample range of 0–400 pg/mL), and 0.028 pg/mL (sample range of 0–428 pg/mL), respectively. Two quality control samples were run at the same time as the samples for each assay. Calibrators and plasma samples, in the case of Simoa, were run in duplicate, and the average of the two measurements was used for statistical analysis. Samples with coefficients of variation higher than 20% were excluded. The investigators involved in the sample analyses were blinded to the clinical diagnosis.

### 4.4. Statistical Analysis

One-way ANOVA and chi-square tests were used for analysis of quantitative and qualitative variables, respectively. The quantitative variables are presented as medians (25th percentile; 75th percentile), and the qualitative variables are presented as percentages. To evaluate the correlation between methods, we used Pearson’s correlation coefficient (r), paired t tests for paired samples, and the Bland–Altman plot. The diagnostic accuracy of the biomarkers/ATN classification was analysed using a binary logistic regression model. In this model, the sensitivity was defined as the percentage of correct diagnoses of AD, and the specificity was defined as the percentage of correct diagnoses of non-AD dementia. The receiver operating characteristic (ROC) curve was further analysed for diagnostic accuracy using the Hanley and McNeil methods [30] to compare the area under the curve (AUC). Values of |z| ≥ 1.96 were considered evidence that the true ROC areas were different. We also performed ROC analysis to determine the cut-off values for the core plasma AD biomarkers and the ratios that best distinguished individuals positive for CSF Aβ42/40. We determined the sensitivity and specificity, and the single analyte value (or ratio) with the highest Youden index (sensitivity + specificity − 1) was identified as the cut-off value. All the statistical analyses were performed using IBM SPSS version 25 (Armonk, NY, USA).

## Figures and Tables

**Figure 1 ijms-25-04594-f001:**
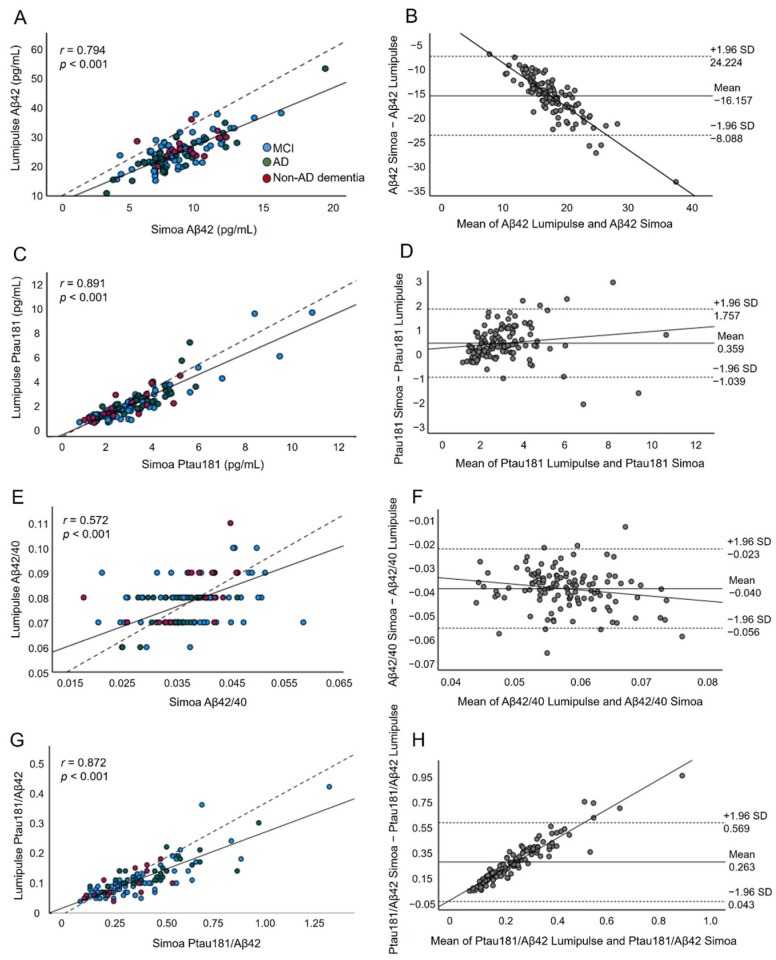
The correlation and Bland–Altman plots for the Aβ42 (**A**,**B**), Ptau181 (**C**,**D**), Aβ42/40 (**E**,**F**), and Ptau181/Aβ42 (**G**,**H**) measurements obtained by the Lumipulse and Simoa methods (*n* = 127). Each point was defined as the Lumipulse and Simoa assay measurements for the same biological sample; in the correlation plots, the blue, green, and red dots represent MCI, AD, and non-AD dementia subjects, respectively. The solid lines represent the estimated regression line, and the dotted line represents the identity line (x = y); in the Bland–Altman plots, the solid lines represent the slopes observed.

**Figure 2 ijms-25-04594-f002:**
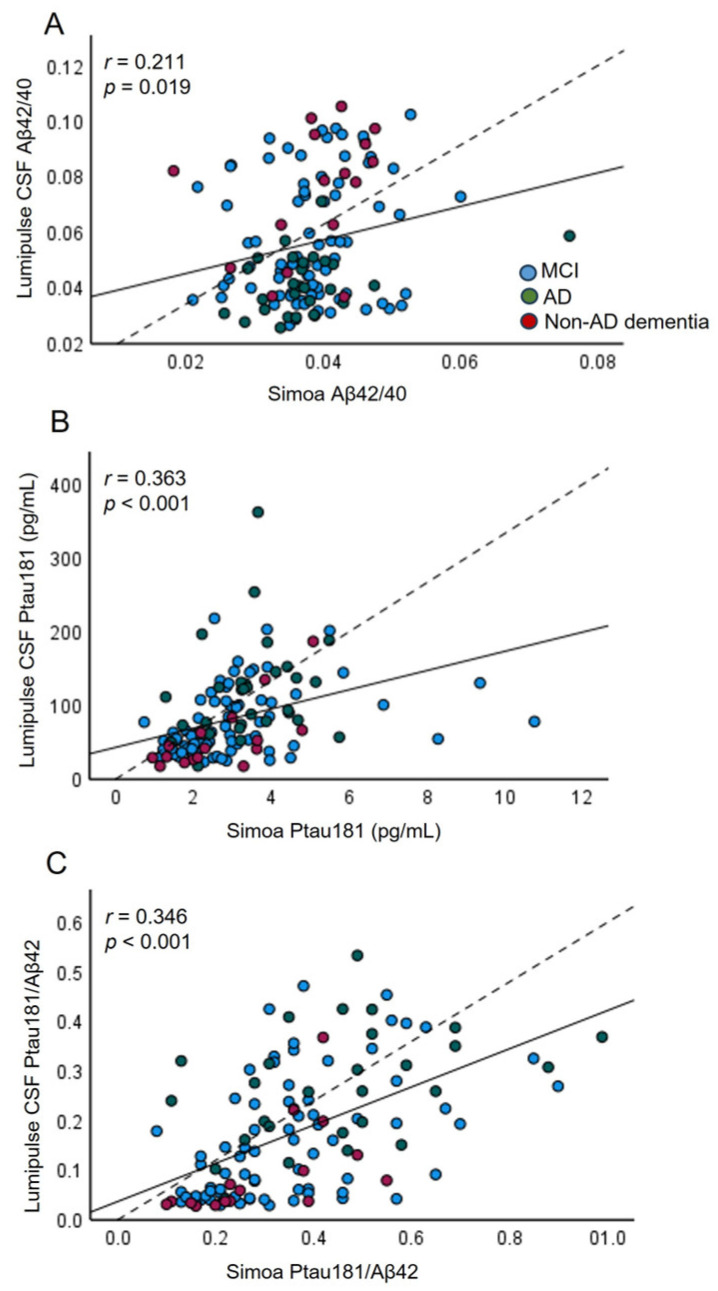
Correlation plots for CSF (Lumipulse) and plasma (Simoa) measurements of Aβ42/40 (**A**), Ptau181 (**B**), and Ptau181/Aβ42 (**C**) (*n* = 127). Each point was defined as the measurement for the same marker detected in CSF and plasma from the same biological sample; the solid lines represent the estimated regression line, and the dotted line represents the identity line (x = y).

**Figure 3 ijms-25-04594-f003:**
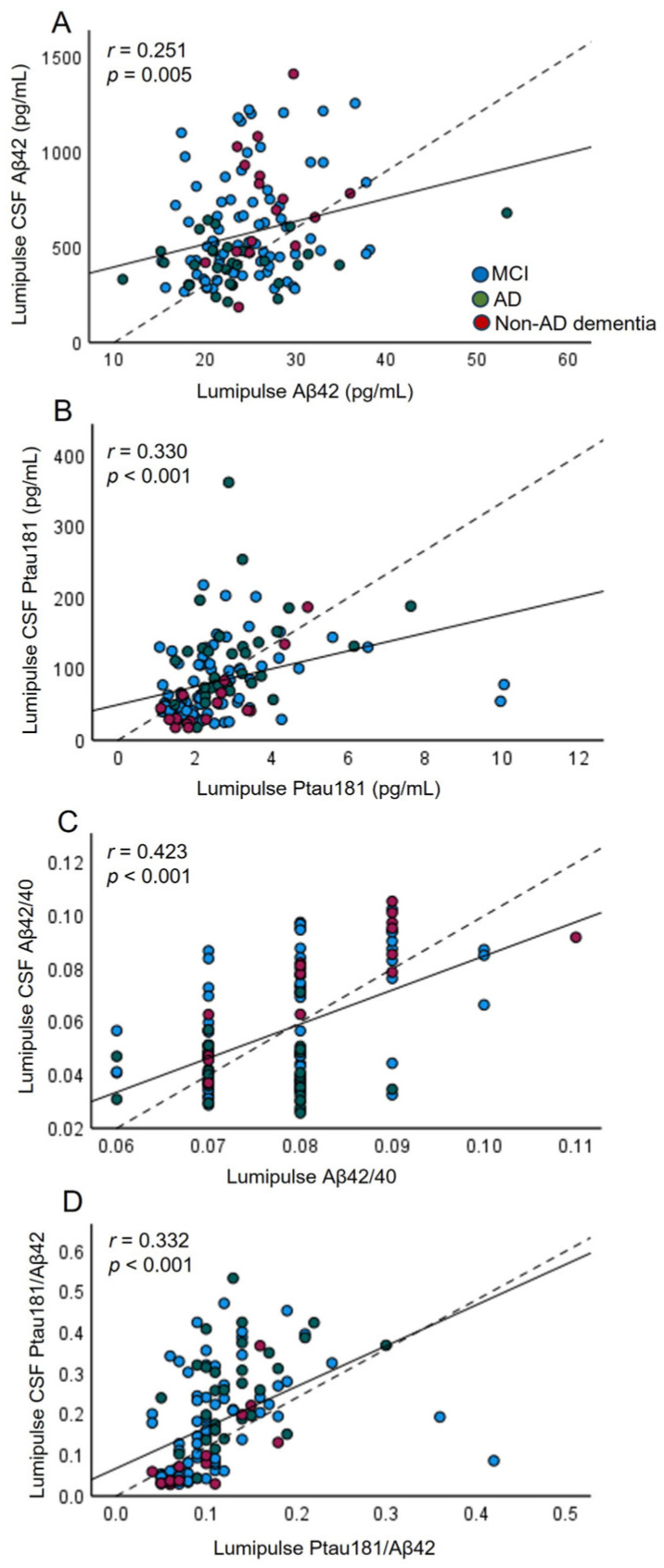
Correlation plots for CSF (Lumipulse) and plasma (Lumipulse) measurements of Aβ42 (**A**), Ptau181 (**B**), Aβ42/40 (**C**), and Ptau181/Aβ42 (**D**) (*n* = 127). Each point was defined as the measurement for the same marker detected in CSF and plasma from the same biological sample; the solid lines represent the estimated regression line, and the dotted line represents the identity line (x = y).

**Figure 4 ijms-25-04594-f004:**
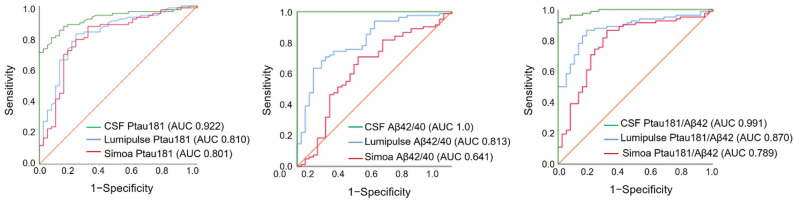
Biomarkers that yielded the maximum Youden index versus Aβ42/Aβ40 ratio status in the receiver operating characteristic analysis.

**Table 1 ijms-25-04594-t001:** The demographic characteristics and biomarker results for all participants and for AD, MCI, and non-AD dementia patients.

	All Participants	MCI	AD	Non-AD Dementia	*p* Value
*n* (%)	127 (100%)	81 (63.7%)	30 (23.6%)	16 (12.6%)	
**Demographic data**					
Age (years)	75 [71;78]	74 [71;77.5]	75.5 [70.5;78]	75.5 [70;78.75]	0.746
Sex (% female), *n* (%)	71 (55.9%)	45 (55.5%)	20 (66.6%)	6 (37.5%)	0.164
Education (years)	11 [8;14]	12 [9;14.5]	11.5 [8;14]	8 [7;11.5]	0.066
Family history of cognitive impairment (yes), *n* (%)	36 (28.3%)	25 (30.8%)	7 (23.3%)	4 (25%)	0.700
**Comorbidities**					
Hypertension, *n* (%)	71 (55.9%)	43 (53.1%)	19 (63.3%)	9 (56.3%)	0.627
Diabetes Mellitus, *n* (%)	28 (22.0%)	17 (21.0%)	8 (26.7%)	3 (18.8%)	0.769
Dyslipidaemia, *n* (%)	56 (44.1%)	37 (45.7%)	13 (43.3%)	6 (37.5%)	0.830
Depression, *n* (%)	45 (35.4%)	29 (35.8%)	9 (30.0%)	7 (43.8%)	0.645
**Lumipulse CSF**					
Aβ42 pg/mL	481 [370;722]	487 [365;752]	412 [309;481]	725 [485;918]	<0.001
Aβ40 pg/mL	10,038 [8142;13,057]	10,052 [8177;13,099]	9900 [7812;12,468]	9196 [8105;12,724]	0.659
Ttau pg/mL	425 [248;718]	412 [232;589]	670 [437;894]	285 [156;493]	<0.001
Ptau181 pg/mL	65.3 [41.2;117.9]	58 [39;104]	116 [75;136]	41 [27;65]	<0.001
Aβ42/40	0.048 [0.037;0.074]	0.048 [0.037;0.076]	0.040 [0.031;0.049]	0.080 [0.051;0.094]	<0.001
Ptau181/Aβ42	0.151 [0.055;0.272]	0.128 [0.051;0.243]	0.281 [0.179;0.373]	0.048 [0.035;0.123]	<0.001
Ttau/Aβ42	0.935 [0.354;1.676]	0.705 [0.319;1.537]	1.655 [1.058;2.203]	0.348 [0.200;0.943]	<0.001
**Lumipulse plasma**					
Aβ42 pg/mL	24 [21;27]	24 [20.9;27.1]	22.5 [19.6;26.1]	25.9 [23.9;29.5]	0.040
Aβ40 pg/mL	303 [276;359]	306 [273;355]	282 [264;339]	320 [281;372]	0.211
Ptau181 pg/mL	2.295 [1.69;3.13]	2.18 [1.64;2.84]	2.86 [2.26;3.45]	2.095 [1.572;3.23]	0.009
Aβ42/40	0.08 [0.07;0.08]	0.08 [0.07;0.08]	0.08 [0.07;0.08]	0.08 [0.07;0.09]	0.093
Ptau181/Aβ42	0.1 [0.07;0.14]	0.09 [0.07;0.12]	0.125 [0.102;0.157]	0.07 [0.06;0.132]	<0.001
**Simoa plasma**					
Aβ42 pg/mL	8.16 [7.07;9.73]	8.00 [6.98;9.26]	7.61 [7.02;9.58]	9.57 [8.23;10.43]	0.111
Aβ40 pg/mL	216 [191;258]	216 [185;257]	212 [193;247]	231 [209;280]	0.271
Ttau pg/mL	2.984 [2.126;3.770]	3.02 [2.32;3.69]	2.81 [1.70;3.90]	3.04 [2.05;3.75]	0.982
Ptau181 pg/mL	2.856 [1.99;3.687]	2.55 [1.96;3.43]	3.36 [2.47;4.34]	2.24 [1.47;3.64]	0.011
Aβ42/40	0.038 [0.034;0.042]	0.038 [0.034;0.042]	0.036 [0.033;0.039]	0.040 [0.034;0.044]	0.254
Ptau181/Aβ42	0.35 [0.22;0.49]	0.32 [0.22;0.435]	0.48 [0.31;0.565]	0.24 [0.17;0.412]	0.010
MMSE score	25 [21;27]	26 [24;27]	20 [17;23.5]	23.5 [17.5;26.5]	<0.001
*APOE4*, *n* (%)	37 (29.1%)	22 (27.1%)	15 (50%)	0 (0%)	0.001

Unless otherwise specified, results are presented as median [IQR]. MMSE, Mini-mental state examination; AD, Alzheimer’s disease; MCI, mild cognitive impairment; non-AD dementia, non-Alzheimer’s disease dementia. *p*-values were calculated by comparing AD, MCI, and non-AD dementia participants using one way ANOVA for continuous variables and Pearson Chi2 for categorical variables.

**Table 2 ijms-25-04594-t002:** Biomarkers with the best discriminating power between AD and non-AD dementia patients.

	Biomarker	AUC (95% CI)	Sensitivity	Specificity	Total % of Predictive Accuracy *
Lumipulse CSF	Aβ42/40	0.879 (0.766–0.992)	87.9%	75.0%	89.1%
Simoa plasma	Aβ42	0.657 (0.493–0.821)	72.4%	56.3%	66.7%
	Aβ40	0.634 (0.466–0.802)	67.9%	43.8%	59.1%
	Ptau181	0.688 (0.516–0.859)	73.3%	56.3%	67.4%
	Ttau	0.511 (0.336–0.685)	75.9%	12.5%	53.3%
	Aβ42/40	0.647 (0.458–0.836)	78.6%	56.3%	70.5%
	**Ptau181/Aβ42**	**0.739 (0.592–0.887)**	**82.8%**	**56.3%**	**73.3%**
	Ttau/Aβ42	0.547 (0.373–0.722)	82.8%	12.5%	57.8%
Lumipulse plasma	**Aβ42**	**0.735 (0.589–0.882)**	**73.3%**	**56.3%**	**67.4%**
	Aβ40	0.662 (0.500–0.823)	72.4%	43.8%	62.2%
	Ptau181	0.664 (0.486–0.841)	70.0%	56.3%	65.3%
	Aβ42/40	0.675 (0.493–0.856)	69.0%	62.5%	66.7%
	**Ptau181/Aβ42**	**0.733 (0.567–0.900)**	**76.7%**	**56.3%**	**69.6%**

AUC, area under the curve. * The percentage of correct classification of AD + correct classification of non-AD/all cases. Biomarkers with the best diagnostic accuracy have been shown in bold.

**Table 3 ijms-25-04594-t003:** Cut-off values for CSF and plasma biomarkers that yielded maximum Youden index versus CSF Aβ42/40 ratio status according to receiver operating characteristic analysis.

		AUC (95% CI)	Sensitivity	Specificity	Max Youden Index	Cut-Off	OPA	Manufacturer Cut-Offs
Lumipulse CSF	Aβ42	0.911 (0.856–0.965)	90.9%	79.5%	70.4%	≤654	87.4%	<600
	Ptau181	0.922 (0.876–0.968)	79.5%	92.3%	71.9%	≥56.15	83.4%	>56.5
	Ttau	0.870 (0.801–0.939)	76.1%	89.7%	65.9%	≥387	80.4%	>400
	Aβ42/40	1.000 (1.000–1.000)	100.0%	100.0%	100.0%	≤0.070	100.0%	<0.069
	Ptau181/Aβ42	0.991 (0.980–1.000)	92.0%	100.0%	92.0%	≥0.091	95.9%	-
	Ttau/Aβ42	0.968 (0.923–1.000)	95.5%	94.9%	90.3%	≥0.517	95.3%	-
Simoa plasma	Aβ42	0.539 (0.429–0.650)	56.5%	38.5%	18.0%	≤8.173	58.1%	
	Ptau181	0.801 (0.712–0.890)	87.2%	69.2%	56.4%	≥2.127	81.6%	
	Ttau	0.505 (0.387–0.622)	91.8%	23.1%	14.8%	≥1.443	70.2%	
	Aβ42/40	0.641 (0.530–0.752)	71.4%	61.5%	33.0%	≤0.039	63.7%	
	Ptau181/Aβ42	0.789 (0.699–0.879)	82.1%	71.8%	53.9%	≥0.270	78.9%	
	Ttau/Aβ42	0.535 (0.422–0.648)	83.5%	33.0%	16.9%	≥0.215	67.7%	
Lumipulse plasma	Aβ42	0.652 (0.551–0.753)	39.5%	92.1%	31.6%	≤21.475	55.6%	
	Ptau181	0.810 (0.727–0.893)	80.5%	76.9%	57.4%	≥2.070	79.4%	
	Aβ42/40	0.813 (0.731–0.895)	64.7%	89.5%	54.2%	≤0.076	71.5%	
	Ptau181/Aβ42	0.870 (0.806–0.934)	86.0%	81.6%	67.6%	≥0.084	84.7%	

AUC, area under the curve; Max Youden index, (sensitivity + specificity − 1); OPA, overall percent agreement.

**Table 4 ijms-25-04594-t004:** Diagnostic accuracy of the ATN classification for CSF and plasma AD biomarkers measured by Lumipulse and Simoa, respectively.

	ATN	AUC (95% CI)	Sensitivity	Specificity	Total % of Predictive Accuracy *	z Value **
Lumipulse CSF	Aβ42/40, Ptau181, Ttau	0.802 (0.639–0.965)	83.3%	75.0%	80.4%	z = −0.617 vs. Simoa Aβ42, Ptau181, (Ttau); z = −0.632 vs. Simoa Aβ42/40, Ptau181, (Ttau)
	Aβ42, Ptau181, Ttau	0.772 (0.590–0.954)	90.0%	68.8%	82.6%	z = −0.350 vs. Simoa Aβ42, Ptau181, (Ttau); z = −0.300 vs. Simoa Aβ42/40, Ptau181, (Ttau)
Simoa plasma	Aβ42/40, Ptau181, Ttau	0.733 (0.576–0.890)	78.6%	56.3%	70.5%	
	Aβ42, Ptau181, Ttau	0.726 (0.573–0.880)	62.1%	81.3%	68.9%	z = −0.065 Simoa Aβ42/40, Ptau181, (Ttau)

AUC, area under the curve. * The percentage of correct classification of AD + correct classification of non-AD/all cases. ** Values of |z| > 1.96 were taken as evidence that the true ROC areas were different.

## Data Availability

The data reported in this manuscript are available within the article and/or its Appendix A. Additional data will be shared upon request by any qualified investigator.

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
