# Peer review of "Assessment of the Correlation and Diagnostic Accuracy between Cerebrospinal Fluid and Plasma Alzheimer’s Disease Biomarkers: A Comparison of the Lumipulse and Simoa Platforms"

_ijms, 2024, doi:10.3390/ijms25094594_

Round 1
Reviewer 1 Report
Comments and Suggestions for Authors
This study presents an intriguing comparison between two platforms, LUMIPULSE and Simoa, for assessing plasma biomarker efficiency. Particularly noteworthy is the comparison with conventional CSF biomarkers. While the authors observed superior performance of CSF biomarkers over plasma, they also conscientiously outlined the study's limitations.
However, several aspects require further refinement:
1. Table 1: The number "24" next to "LUMIPULSE Plasma" should be removed.
2. In the sentence, "0.572 for Aβ42/40 (p < 0.001) (Figure 1E) and 0.837 for 108 Ptau181/Aβ42 (p < 0.001) (Figure 1G)," it is stated that "0.572 for Aβ42/40" indicates a high correlation. However, this value does not signify a high correlation. Please revise for clarity.
3. Ensure consistency in referring to pTau181 in Figure 1 and align it with the text. Instead of pTau use pTau181 in all figures.
4. Check the fonts in Figure 1 description and particularly the font "p181” in the entire manuscript
5. In Figure 1F, is there a mild negative trend? given the value of -0.04, indicating almost no difference.
6. Clarify why Simoa was not employed for AD CSF markers.
7. Include R and p values within correlation plots for better understanding.
8. Differentiate groups with various colours in correlation plots (for all 127 points) to visualize trends effectively.
9. In Table 2, there are no values for CSF pTau and pTau/Aβ42. Please address this inconsistency.
10. The title of Table 4 is deemed inappropriate as it contains both CSF and plasma samples. Consider revising the title accordingly.
Author Response
This study presents an intriguing comparison between two platforms, LUMIPULSE and Simoa, for assessing plasma biomarker efficiency. Particularly noteworthy is the comparison with conventional CSF biomarkers. While the authors observed superior performance of CSF biomarkers over plasma, they also conscientiously outlined the study's limitations.
However, several aspects require further refinement:
- Table 1: The number "24" next to "LUMIPULSE Plasma" should be removed.
R/ The number 24 that seems to stand alone, is the median of Aβ42 for the MCI subgroup. It appears in an upper line because the interquartile range was presented with 3 decimal digits, so we eliminated 2 digits from the 25th and 75th interquartile range. In addition, we noticed that some information in the tables have been displaced after converting our original document to the format of the journal. We have tried to order a little bit the tables but surly the journal editors will take it in into account before publishing the final version.
- In the sentence, "0.572 for Aβ42/40 (p < 0.001) (Figure 1E) and 0.837 for 108 Ptau181/Aβ42 (p < 0.001) (Figure 1G)," it is stated that "0.572 for Aβ42/40" indicates a high correlation. However, this value does not signify a high correlation. Please revise for clarity.
R/Thank you for the comment. We corrected the sentence (lines 117-119).
- Ensure consistency in referring to pTau181 in Figure 1 and align it with the text. Instead of pTau use pTau181 in all figures.
R/ All figures were modified.
- Check the fonts in Figure 1 description and particularly the font "p181” in the entire manuscript.
R/Thank you for the comment. The changes in the font throughout the manuscript were probably produced when our original manuscript was formatted according to the IJMS by the editors of the journal as they did not exist in our submitted manuscript. However, we modified all detected font inconsistencies.
- In Figure 1F, is there a mild negative trend? given the value of -0.04, indicating almost no difference.
R/ Please take into consideration that in the case of Ab42/40 ratio, the values are small (approximately 0.06-0.1 for Lumipulse and 0.02-0.05 for Simoa (Figure 1E)). Therefore, when the overall differences regarding Ab42/40 ratio between two methods are small (-0.023 to -0.056, figure 1F) a mean difference of -0.04, will generate the negative trend. This negative trend can also be seen by comparing the Ab42/40 range of values for each method (Figure 1E).
- Clarify why Simoa was not employed for AD CSF markers.
R/ We measured the CSF biomarkers by Lumipulse as our hospital is equipped with a Lumipulse G and determination of CSF AD biomarkers by this platform is part of the routine clinical practice of our memory clinic. Therefore, we had the CSF AD biomarker determinations by Lumipulse as a part of clinical practice and then, we decided to compare SIMOA and Lumipulse regarding plasma biomarkers. This information was added to the Lines 280-282 of the discussion section. We agree with the reviewer that it would be interesting to have the results of the CSF biomarkers by Simoa platform too but due to budget and sample volume limitations, this was not possible. We accept that it is one of the limitations of our study as mentioned in the limitation section (Lines 335 and 336).
- Include R and p values within correlation plots for better understanding.
R/ The authors are grateful for the comment. The R and p-values were introduced to all correlation plots.
- Differentiate groups with various colours in correlation plots (for all 127 points) to visualize trends effectively.
R/ The authors appreciate this comment. We substituted all the correlation plots by new ones, in which the groups are separated by colours.
- In Table 2, there are no values for CSF pTau and pTau/Aβ42. Please address this inconsistency.
R/ In Table 2, we only included the best CSF AD biomarker for discriminating the diagnostic groups (AD vs non-AD dementia) because our objective was not to determine the diagnostic accuracy of all CSF AD biomarkers as was performed in our previously published work (PMID: 33746733). Our objective was to compare the diagnostic accuracy of plasma biomarkers determined by Simoa and Limupulse with the CSF AD biomarker with the best discriminating power. However, the reviewer is completely right as it was not defined well in our objective section. We modified the objective c (line 87) to prevent inconsistency between results and objectives and clarified our objective in the results section too (Lines 184-185).
- The title of Table 4 is deemed inappropriate as it contains both CSF and plasma samples. Consider revising the title accordingly.
R/ The title of Table 4 was corrected.

Reviewer 2 Report
Comments and Suggestions for Authors
This study is well-conceived and straightforward, I have only few minor comments.
- The writing needs to be revised to make it easier to follow for the reader. There are many long and complex sentences, with twisted grammar. Few examples on lines 46-53, 78-80250-253, ...
- the detection limit for the Lumipulse should be defined in the intro, as done for the Simoa.
- what is the "best agreement" on line 82?
- optional comment: the use of ATN classes in section 2.6 is somewhat confusing and leaves the reader clueless if he is not familiar with this classification.
Comments on the Quality of English Languagesee comments to authors.
Author Response
This study is well-conceived and straightforward, I have only few minor comments.
- The writing needs to be revised to make it easier to follow for the reader. There are many long and complex sentences, with twisted grammar. Few examples on lines 46-53, 78-80250-253, ...
R/ Thank you for the comment. Our manuscript was edited for correct English language, grammar, punctuation, and phrasing by American Journal Experts that is part of Springer Nature. We attach the certification of the edition. However, if you think that the manuscript needs more editing, we will be pleased to ask AJE for an additional English editing.
- the detection limit for the Lumipulse should be defined in the intro, as done for the Simoa.
R/ The detection limit for Lumipulse was added to the introduction (Lines 72-73).
- what is the "best agreement" on line 82?
R/ The best agreement was the value that optimized the concordance with the CSF Aβ42/40-positive and Aβ42/40-negative status. In other words, the value with the best discriminating accuracy between the CSF Aβ42/40-positive and Aβ42/40-negative status. To clarify the objective “d”, we changed the sentence from “to determine the cut-offs of plasma AD biomarkers measured by Simoa and Lumipulse based on the best agreement with the CSF Aβ42/40 status” TO “to determine the cut-offs of plasma AD biomarkers measured by Simoa and Lumipulse based on the best discriminating accuracy between the CSF Aβ42/40 positive and negative status” (Lines 88-89).
- optional comment: the use of ATN classes in section 2.6 is somewhat confusing and leaves the reader clueless if he is not familiar with this classification.
R/ Although the ATN classification is not used for the clinical diagnosis of AD, it classifies the patients based on the presence or absence of the biological evidence of the AD-related pathology (Amyloid, p-tau, and neurodegeneration). This classification, after its introduction by NIA-AA in 2018 (PMID: 29653606), is recommended to be used for stratification of patients in the continuum of AD for investigation and clinical trials. Since then, it has been used in many research studies on AD biomarkers (e.g. PMID: 32522798; PMID: 33811742; PMID: 33811742; and PMID: 36130840). So, we ask the reviewer to let us to keep the section 2.6 with the current format.

Reviewer 3 Report
Comments and Suggestions for Authors
The main objective of the article developed by Farida Dakterzada and collaborators was to evaluate the correlation and agreement between the plasma biomarkers of AD measured with Lumipulse and Simoa. In general, the manuscript is well-written and structured, however, to be published, it is necessary to follow the recommendations.
Major concerns
1. My greatest concern is the Non-AD subjects, given that a significant percentage present comorbidities, and in the MMSE score they present a score compatible with MCI. This is a great bias since these could cause modifications in the measurements made. I suggest increasing the number of control subjects, as well as delimiting the exclusion criteria and avoiding including subjects with MCI. Additionally, the clinical evaluation could be carried out with a more specific and sensitive method, for example MoCA test.
2. In methods, the demographic variables studied need to be described. What were the exclusion criteria? It is not clear how long the patients have progressed or the time and type of treatments used.
3. Why were CSF measurements not performed through SIMOA?
4. From a statistical point of view, is it valid to compare such disproportionate populations?
Minor concerns
1. In the introduction, include the epidemiology of AD and highlight the importance of pTau181
2. The justification for comparing Lumipulse and Simoa Platforms needs to be further strengthened.
3. In the graphs add the correlation coefficient and the p value obtained. It remains to be specified in the graphs that the variable evaluated was pTau 181.
4. Verify the homogeneity of the data in the values obtained in AB42/40 plasma through Lumipulse. Avoid rounding values.
5. In the footer of the table, define what the letters in bold mean.
6. Increase the size and quality of Figure 4.
7. Order your tables better since you do not identify where each row begins and ends.
Comments on the Quality of English LanguageMinor editing of English language required
Round 2
Reviewer 3 Report
Comments and Suggestions for Authors
I congratulate the authors, good work. The only confusing thing is the term non-AD dementia patients, changing it so that it is understood that they are patients with another type of dementia.
Author Response
I congratulate the authors, good work. The only confusing thing is the term non-AD dementia patients, changing it so that it is understood that they are patients with another type of dementia.
R/ We thank the reviewer for his/her time and effort in reviewing our manuscript. As AD is the most common type of dementia (60-70%), the term non-AD dementia usually is used to refer to different types of dementia other than AD. In the methods section, we added the types of dementia that were included in non-AD dementia group (lines 354-357) to clarify better the patients that were included in this group.